# Proton Irradiation Effects on Hardness and the Volta Potential of Welding 308L Duplex Stainless Steel

**DOI:** 10.3390/mi10010011

**Published:** 2018-12-25

**Authors:** Baolong Jiang, Qunjia Peng, Zhijie Jiao, Alex A. Volinsky, Lijie Qiao

**Affiliations:** 1Beijing Advanced Innovation Center for Material Genetic Engineering, Key Laboratory for Environmental Fracture (MOE), University of Science and Technology Beijing, Beijing 100083, China; jiangbaolong@xs.ustb.edu.cn (B.J.); volinsky@usf.edu (A.A.V.); 2Key Laboratory of Nuclear Materials and Safety Assessment, Institute of Metal Research, Chinese Academy of Sciences, Shenyang 110016, China; pengqunjia@cgnpc.com.cn; 3Nuclear Engineering and Radiological Sciences, University of Michigan, 2355 Bonisteel Boulevard, Ann Arbor, MI 48109, USA; zjiao@umich.edu; 4Department of Mechanical Engineering, University of South Florida, Tampa, FL 33620, USA

**Keywords:** 308L, proton irradiation, duplex microstructure, Volta potential, micro-hardness

## Abstract

308L welding duplex stainless steel has been irradiated at 360 °C with 2 MeV protons, corresponding to a dose of 3 dpa at the maximum depth of 20 μm. Microhardness of the δ-ferrite and austenite phases was studied before and after proton irradiation using in situ nanomechanical test system (ISNTS). The locations of the phases for indentations placement were obtained by scanning probe microscopy from the ISNTS. The hardness of the δ-ferrite had a close relationship with the vacancy distribution obtained from the Stopping and Range of Ions in Matter (SRIM) Monte Carlo simulation code. However, the hardness of the austenite phase in the maximum damage region (17–20 μm depth) from the SRIM simulation was decreasing sharply, and a hardness transition region (>20 μm and <55 μm depth) was found between the maximum damage region (17–20 μm depth) and the unirradiated region (>20 μm depth). However, the δ-ferrite hardness behavior was different. A hardness of the two phases increased on the irradiated surface and the interior due to different hardening mechanisms in the austenite and δ-ferrite phases after a long time high-temperature irradiation. A transition region (>20 μm and <55 μm depth) of the Volta potential was also found, which was caused by the deeper transfer of implanted protons measured by scanning Kelvin probe force microscopy.

## 1. Introduction

Many devices are irradiated by various high-energy ions in the primary circuit environment of pressurized water reactors (PWR) [1,2,3]. Ionic bombardment can produce many defects in metals, and irradiation-induced material damage accumulates with prolonged equipment use [4]. One form of resulting damage is irradiation hardening. When metal is hardened by ion irradiation, it becomes more brittle. Therefore, some components, such as springs and fasteners, could break, causing damage to larger parts if not replaced [5,6]. Irradiation-assisted stress corrosion cracking is also a concerning problem [7,8,9]. Additionally, irradiation hardening could produce local stress concentrations, which greatly enhance the stress corrosion cracking. At present, various studies of irradiation hardening have been conducted using austenite stainless steels (face-centered cubic, fcc), such as 304L and 316L, which are widely used as nuclear structural materials [10,11,12]. In addition, other nuclear materials, including oxide dispersion strengthened steels (body-centered cubic, bcc) [13,14], vanadium alloys [15], high-entropy alloys and reactor pressure vessels (RPV) steels have also been studied [16,17,18,19,20]. Many factors, which affect irradiation hardness have been considered, including irradiation dose, temperature, and defect types. However, defect types are most important, as the temperature of PWR is 300–400 °C in different circuits and hardening tends to saturate at some radiation doses [21,22,23]. It can be concluded that irradiation hardening has a close relationship with defect types formed in the bulk when the irradiation dose is below saturation. 

308L stainless steel welded overlay cladding is used on the surface of RPV to protect it from corrosion in a water environment. It has approximately 10% δ-ferrite phase (bcc) and 90% austenite phase (fcc), which are subjected to strong ion irradiation from the reactor core [24]. Takeuchi et al. [24,25] investigated the effects of thermal aging and neutron irradiation on microstructure and hardness of 308L steel using atom probe tomography and nanoindentation testing. They found that the primary factors responsible for the hardening of the δ-ferrite phase are Cr spinodal decomposition and NiSiMn clusters formed by thermal aging [26,27]. Furthermore, the hardening of the δ-ferrite phase occurred due to both irradiation and aging, since the Cr concentration in the δ-ferrite phase is enhanced by irradiation [27]. However, the NiSi clusters formed by irradiation only cause hardening of the austenite phase [28].

It is also noticeable that hardening mechanisms of ferrite and austenite are different according to the Hall-Petch formula σ_y_ = σ_0_ + K_y_·d^−1/2^ [29] due to the relationship between the lattice frictional resistance (σ_0_), dislocation pinning force (K_y_), yield strength (σ_y_), and radiation dose. In bcc metals, only σ_0_ changes significantly with increasing radiation dose. Therefore, lattice friction hardening mechanisms are mainly found in irradiated bcc materials, while fcc materials collectively suffer from lattice friction hardening and source hardening, since both σ_0_ and K_y_ are affected by the irradiation dose. Chen et al. [30,31] studied the defects formed after irradiation of the cast austenitic stainless steel (CASS) CF8 cast stainless steel, which has both austenite and δ-ferrite phases. It has been observed that different kinds of defects formed in the material when various phase structures were irradiated by heavy ions. At a high dose of three displacements per atom (dpa), a dislocation network formed in the fcc structure, while the bcc structure exhibited extended dislocations as line segments. One could make a preliminary judgment that the irradiation hardening of different structural materials has a strong relationship with defect types formed in them, and the hardening mechanisms can be changed with different irradiation doses.

Thus, most studies of irradiation hardening are focused on single phase steels, with less emphasis on duplex steels. Some studies considered uniform damage region of the duplex steels [32]. The surface is exposed to the environment and the maximum damage region has many more dpa than the uniform damage region, which could simulate the 40 years operation under irradiation conditions [33]. The surface and the maximum damage regions along the depth direction should also be considered and studied systematically in addition to the uniform damage region. Most irradiation experiments were carried out using protons substitutes for neutrons because the radiation damage effects caused by neutrons are close to protons [34]. However, when metal is implanted by high energy protons, a transition region of protons exists in the unirradiated region exceeding the irradiated depth and the specific depth of this region has not been determined by any effective means [35,36].

Scanning Kelvin probe force microscopy (SKPFM) is an effective method to detect hydrogen by mapping the contact potential difference (CPD) changes and has been increasingly used in corrosion studies [37]. Guo et al. observed the hydrogen-induced CPD changes in duplex stainless steel (DSS) due to different hydrogen solubility and investigated DSS pitting corrosion behavior using SKPFM [38]. In addition, Stratmann et al. also reported that CPD had a linear relation with the corrosion potential at the metal-solution interface [39,40]. SKPFM can directly measure the change of hydrogen concentration. Therefore, the unirradiated region along the depth of the irradiated sample should also be studied by SKPFM.

In this paper, 308L stainless steel has been studied. 308L stainless steel is used as a welding material in the primary loop of pressurized water reactors because it exhibits enhanced corrosion resistance due to its duplex phase structure [41,42]. Nanoindentation was used to investigate hardening of austenite and δ-ferrite caused by proton irradiation. Surface, uniform, maximum damage, and unirradiated regions along the depth direction of the irradiated sample were used for hardness testing. In addition, the transition region of protons was directly measured by SKPFM in the unirradiated region past the irradiated depth. Thus, the specific depth of the transition region was obtained.

## 2. Materials and Methods

The material used for the experiments is 308L duplex stainless steel. The hard-facing layer was cut from a mockup of a safe-end weld with a nominal chemical composition of 68% Fe, 19.88% Cr, 10.3% Ni, 1.33% Mn, 0.32% Si, 0.065% Cu, 0.016% C, 0.015% P, 0.011% S, and 0.01% Nb, in wt%. The samples were in the form of rectangular bars with 20 × 3 × 2 mm size, which were cut from the weld joint for proton irradiation. The upper-end surface of the sample before irradiation was mechanically ground by silicon carbide paper up to 5000 grit, then polished using 0.5 μm diamond paste, and finally polished by 0.04 μm colloidal silica slurry. Finally, the stress layer was removed from the surface, and the samples were ultrasonically cleaned in ethanol. Pure nickel used for calibration as a standard sample was polished up to 2000 grit with SiC paper in water and ultrasonically cleaned in distilled water.

Proton irradiation was performed using a custom-designed stage connected to the General Ionex Tandetron accelerator at the Michigan Ion Beam Laboratory. Irradiation experiments were conducted using 2 MeV protons at a dose rate of approximately 6 × 10^−6^ dpa/s, resulting in nearly uniform damage throughout the first 15 μm of the 20 μm proton depth range. The experimental doses and dose rates were calculated using the stopping and range of ions in matter (SRIM 2008, formerly Transport of Ions in Mater, TRIM) Monte Carlo simulation code. The displacements per atom were calculated using the SRIM with a displacement energy of 40 eV, as recommended in the ASTM E521-8 [43,44,45]. The irradiated surface area was approximately 60 mm^2^. The specimen temperature was maintained during irradiation at 360 ± 5 °C.

Microhardness was measured by the in situ nanomechanical test system (Hysitron, TI-900 Triboindenter, Minneapolis, MN, USA). Scanning probe microscopy (SPM) imaging and in situ hardness measurements were carried out using a contact mode to record the surface topography with the Berkovich tip. From the topography of the δ-ferrite and austenite phases, the δ-ferrite had a very narrow size of 3–5 μm. Therefore, the 1500 μN maximum load was used to measure hardness and avoid indentation depth in excess of the δ-ferrite width. 

SKPFM is one of the modes of SPM, which can detect the Volta potential of the material. Dimension Nanoscope V (Veeco Instruments Inc., New York, NY, USA) was used for the SKPFM measurements. To avoid environmental effects, the SKPFM measurements were conducted in air at room temperature and relative humidity of about 25%. The tip used in the SKPFM measurements was the NT-MDT conductive TiN-coated silicon tip (Moscow, Russia) with 2.5 N/m force constant and 35 nm tip radius. The lift mode was used to record the Volta potential signal at 60 nm distance from the sample surface [46].

In the SKPFM measurements, AC voltage is applied to biased oscillating conductive tip [47,48]. The topography and the Volta potential were detected by normal atomic force microscopy (AFM) detection scheme. After that, a reverse DC voltage with the same magnitude was applied to the tip to make up the difference of AC voltage and stop the tip oscillation. Finally, the signal was inverted by the output signal from the instrument. In SKPFM, the contact potential difference was calculated as the difference between the tip and the sample voltage: *V*_CPD_ = (*ϕ*_tip_ − *ϕ*_sample_)/*e*, where *ϕ*_tip_ and *ϕ*_sample_ are the work functions of the tip and the sample, and *e* is the electron charge. From the work function of the tip (*ϕ*_tip_), the work function of the sample (*ϕ*_sample_) can be deduced. The work functions have been used for calculating the corrosion rates [49]. Many experimental observations and theoretical studies indicated that the work function has a close relationship with the corrosion potential. The lower the work function of the material, the easier it corrodes, so that the work function is a sensitive parameter reflecting corrosion behavior [50,51]. 

The detected Volta potential values are also influenced by the tip characteristics, such as the metal coating and the probe geometry. In order to avoid the corresponding errors, a calibration experiment of the Volta potential was carried out using a pure Ni surface with the stable known potential [52].

## 3. Results

### 3.1. Proton Irradiation Effects on Hardening of Different Regions 

#### 3.1.1. Irradiation Regions from the SRIM Simulation Curves

The proton distribution and the profile of radiation-induced displacement per atom were generated using the SRIM code for the 2 MeV protons, as shown in Figure 1. The dpa (vacancies produced/atom = displacements per atom) and atomic implantation ion concentration (in at%) were obtained by using Equations (1) and (2) [53]. As shown in Figure 1, four regions were selected for measurements after proton irradiation. One was the surface region I of the irradiated sample, the next was the uniform damage region II, the third was the maximum damage region III of implanted ions and vacancies in the final irradiated depth and the last was the unirradiated region IV deeper in the sample, listed in Table 1. At the initial stage, four measurement regions were obtained from the SRIM simulation curves, while the unirradiated region along the depth direction could be divided into two regions, including a region affected by protons and another region unaffected by irradiation, which would be characterized by nanoindentation and SKPFM in the next section of this study. 

#### 3.1.2. Ferrite and Austenite Microhardness

Figure 2a shows the unirradiated δ-ferrite and austenite topography of the 308L steel measured by SPM. It is seen that the δ-ferrite phase has a much smaller size than the γ-austenite phase and the indentation point could be at the boundary of the two phases due to the thermal drift. In order to avoid thermal drift effects, at least five indentations were performed to calculate the average hardness of each phase. Figure 2b shows representative indentation load-displacement curves of the two unirradiated phases. The hardness of δ-ferrite was higher than austenite at the same maximum load of 1500 μN since austenite had larger indentation depth for the same maximum load [54]. The average hardness for each phase is: unirradiated δ-ferrite 5.62 ± 0.34 GPa and unirradiated austenite 4.09 ± 0.23 GPa.

Figure 3 shows the cross-sectional SEM image of the irradiated sample after colloidal silica slurry polishing. A clear boundary was found in the irradiated and unirradiated regions after polishing with colloidal silica slurry. This line is the irradiation boundary (region III). In addition, the δ-ferrite phase was stretched across the sagged austenite phase in the irradiation boundary after polishing. This result indicated that the δ-ferrite phase was harder than the austenite in this region, also confirmed by the nanoindentation hardness measurements: δ-ferrite 6.8 ± 0.40 GPa > austenite 2.78 ± 0.08 GPa. The austenite in the boundary region III has the lowest hardness. The δ-ferrite in the boundary has the maximum hardness compared with any other region.

#### 3.1.3. SKPFM Volta Potential of the Irradiated Sample Cross-Section

Figure 4a shows the Volta potential changes with depth from 10 μm to 80 μm below the irradiated surface. Figure 4b presents the corresponding Volta potential and height topography maps. It can be observed that the transition region IV of the Volta potential formed in the unirradiated region below the irradiated region. The depth of the Volta potential transition region was approximately 35 μm. In addition, the transition region of hardness has also been found in austenite in the same region of the Volta potential transition, as shown in Figure 4c,d. Hardness values were obtained every 3 μm from the nanoindentation measurements, and at least two indentation points were used for hardness testing at each depth, as shown in Figure 4c. The hardness values of the transition region were obtained by taking the average value of all indentation points in this region, as shown in Figure 4d.

Figure 5 shows the hardness of δ-ferrite and austenite from the irradiated surface region I, uniform damage region II, maximum damage region III, to unirradiated region IV along the depth direction, and the hardness of the two phases in the unirradiated sample. The hardness of the δ-ferrite phase has a very close relationship with the SRIM vacancy distribution. However, the hardness of the austenite phase in the boundary region III was sharply decreasing, and there was a hardness transition region from the boundary region III to the unirradiated region IV. In order to ensure that the test region exceeds the region affected by hydrogen ions, the surface data of the samples hardness without irradiation was compared to the region unaffected by irradiation in the depth direction. It was observed that the region exceeding the transition region depth has the same hardness as the unirradiated surface.

The δ-ferrite did not exhibit the same hardness change as austenite. In addition, the hardness values of δ-ferrite were greater than austenite on the irradiated surface (region I) compared with the unirradiated hardness results. However, the hardness value of δ-ferrite changed more than the austenite in the uniform damage region II. It could be affected by the different irradiation doses in different regions, as shown by the SRIM simulation curves. The δ-ferrite phase was also affected by thermal aging hardening, except the irradiation hardness, which was different from austenite [24,26].

## 4. Discussion

Figure 6 shows how proton irradiation affects the hardness of austenite and δ-ferrite in different regions. It seems that many more vacancies and interstitials would transfer to the surface region I from the near surface of uniform damaged region II in the irradiated sample, as the arrangement of atoms on the metal surface was obviously different from the bulk [55]. Therefore, the vacancy distribution has an increasing trend from 0 to 3 dpa of the uniform damage region II. It appears that when the DSS material was irradiated by protons below 3 dpa, the effect of proton irradiation on austenite hardness was greater than δ-ferrite. Hardness increased at the irradiated surface by 0.43 GPa for the δ-ferrite and by 0.5 GPa for the austenite. In addition, it has been mentioned that irradiation-induced hardening in metals was caused by the production of various defects [56,57]. However, hardening of the δ-ferrite phase was caused by both thermal aging and irradiation due to 360 °C irradiation temperature for 138 h, and only the austenite phase was affected by irradiation [24,25,26]. In many studies [29,58], from the Hall-Petch formula, metals with a bcc structure were mainly affected by friction hardening and the influence of source hardening was less. Source hardening is the increase in stress required to start a dislocation moving on its glide plane [58]. However, for metals with an fcc structure, the effect of hardening occurred simultaneously by the two hardening mechanisms. Chen et al. [30,31] have studied irradiation-induced dislocation loops, which appeared at a much lower dose in the austenite than in the δ-ferrite. Many visible vacancy-type defect clusters have been found directly by transmission electron microscopy (TEM) in Cu and Ni fcc metals. However, these visible defect clusters were not found directly by TEM in Fe bcc metals irradiated by neutrons at low doses [59]. Victoria et al. had also found that a much lower dose was needed for fcc metals than bcc metals to produce the same number of dislocation loops under neutron and proton irradiation [60]. It was suggested that defects were more easily formed in fcc phases than bcc phases, attributed to their different crystal lattice structure. 

In this study, the effect of irradiation hardening of the austenite phase on the irradiated surface region I was greater when the irradiation dose was below 3 dpa. This result was consistent with the dislocation loops and visible vacancy-type defect clusters appearing at a much lower dose in the austenite fcc structure, which were mainly affected by source hardening, as shown in Figure 6I image, where austenite phase has some dislocation loops in the surface region and no dislocation loops in the δ-ferrite phase. At this condition, δ-ferrite phase was mainly affected by thermal aging hardening with a slight influence of irradiation. Although the irradiation ions used by Chen et al. were different from this paper, it is known that heavy ions are efficient at producing denser cascades than protons, and a significant microstructure difference of austenite and δ-ferrite phases could be formed by heavy ions irradiation [58]. Proton irradiation could also produce the microstructure difference, and the microstructure formed in the austenite phase could have dislocation loops or visible vacancy clusters at low proton doses, which needs further investigation by TEM.

The vacancy distribution was 3 dpa in the uniform damage region II, as shown in Figure 6II. Chen et al. [30,31] also found that the type of defects formed in δ-ferrite and austenite phases were different due to the 3 dpa irradiation by heavy ions. In the fcc structure, a dislocation network microstructure was formed, while the bcc structure exhibited an extended dislocation structure in the form of line segments. It could also be deduced that different microstructure would be formed in fcc and bcc structures irradiated by protons, due to their different crystal lattice structures, as mentioned above. The hardness of austenite was lower than δ-ferrite in this region. In this condition, the austenite and δ-ferrite phases were both affected by vacancies [61]. Furthermore, the δ-ferrite phase in this region was also affected by thermal aging [24,26]. Nakata et al. [62] found that voids were formed only in δ-ferrite of the weld metal irradiated at 773 K. In general, cavities and voids were the strongest barriers between all the barriers formed by irradiation [21]. Therefore, the variation of hardening produced in austenite was less than in the δ-ferrite phase due to these reasons.

As seen from Figure 6III and the SRIM curve of implant ions (H^+^) and vacancies distribution on the left of Figure 6, the maximum damage region III has a very large number of H^+^ ions. H^+^ ions could not stay in the bulk and many more H^+^ ions would transfer to outside of the material, leaving fewer H^+^ ions to form hydrogen atoms in the bulk [35,36]. The concentration of hydrogen in the maximum damage region III was one order of magnitude higher than in other regions. Guo et al. [48,52] studied the corrosion properties of dual-phase steel by electrochemical hydrogen charging and found that more hydrogen was dissolved in the austenite phase than the δ-ferrite. Figure 3 shows the δ-ferrite phase along the sunken austenite phase caused by polishing. It could be deduced that a larger amount of H distributed in the austenite phase than the δ-ferrite phase makes it more likely for hydrogen embrittlement to occur in austenite. In addition, the austenite in this region could also soften due to many more voids formed by many accumulated vacancies [63]. It was also found that voids could be more easily formed with hydrogen presence [35]. Additionally, it is also well known that hydrogen diffusivity in different phases is largely different. Hydrogen diffuses in ferrite much faster than in austenite [64]. This may be another possible origin of the absence of hydrogen-induced softening in ferrite. Therefore, when the sample was polished, the austenite phase was first sunk. The hardness of the austenite phase was the lowest of all regions, even smaller than the unirradiated austenite.

The hardness of the austenite has a transition between the maximum damage region III and unirradiated region IV in the cross-section. However, the hardness of δ-ferrite has not exhibited the same results, as shown in Figure 5. It was deduced that the transition region was formed by the influence of the sunken austenite in the maximum damage region III. When the H^+^ ions were concentrated in the maximum damage region III, a region exists where the H^+^ ion concentration declines from this region to the unirradiated region IV. Since the austenite phase could dissolve more hydrogen, and the size of the austenite phase is larger than δ-ferrite, the austenite phase provides a favorable channel for the transmission of hydrogen [50]. It was also found that when materials were irradiated by protons, the depth affected by protons was deeper than the SRIM simulation curves of implanted protons due to many more voids formed as channels for protons transmission [36].

As shown in Figure 6IV, apart from the above results, the austenite phase also has a transition region of the Volta potential in the hardness transition region detected by SKPFM. Guo et al. [50,52] found that the dissolved hydrogen in the austenite phase has a low potential, so it was deduced that the Volta potential transition has a very close relationship with the dissolved hydrogen in austenite. Since the size of δ-ferrite is too small, the influence of δ-ferrite on the Volta potential in the transition region could be negligible. Since the gap size between the two phases was so large, more studies should be conducted to investigate other dual-phase steels, such as 2205 and 2507. Additionally, the influence of dissolved hydrogen needs to be studied in more detail by TEM or other methods.

## 5. Conclusions

The effects of irradiation on the hardness of 308L welding DSS stainless steel were investigated in this study and the following conclusions are drawn.

(1) There is a close relationship between the hardness of the δ-ferrite and the vacancy distribution obtained from the SRIM simulation curves. A different result of austenite has been found with a hardness decrease in the maximum damage region III from the SRIM simulation, and a hardness transition region has also been found between the maximum damage region III and the unirradiated region IV. 

(2) The hardness behavior of δ-ferrite and austenite are different on the irradiated surface and the interior due to their different hardening mechanisms, where austenite is only affected by irradiation hardening and δ-ferrite is affected by irradiation hardening and thermal aging hardening combined.

(3) Due to the size effect and the difference of the solubility of hydrogen in the two phases, a hardness and surface potential transition regions exist in the austenite phase from the maximum damaged region III to the unirradiated region IV of the cross-section, which were affected by hydrogen in this region directly detected by SKPFM.

## Figures and Tables

**Figure 1 micromachines-10-00011-f001:**
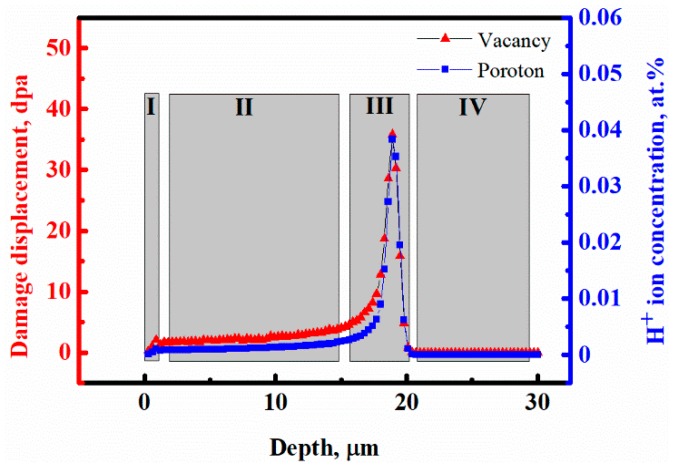
SRIM 2008 simulation of the damage zone and implanted protons distribution: region I at the surface; region II, uniform damage; region III, maximum damage; and IV, unirradiated region along the sample depth.

**Figure 2 micromachines-10-00011-f002:**
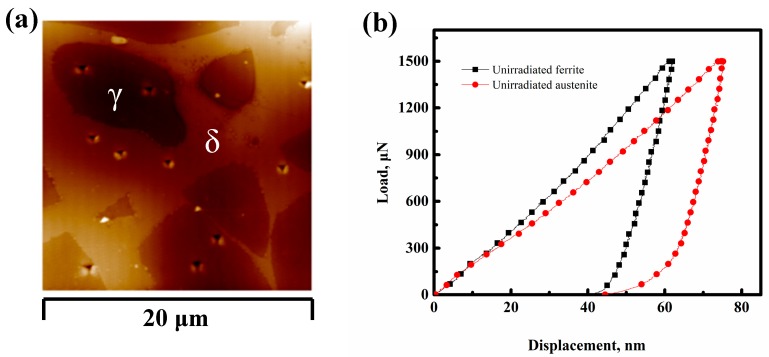
(**a**) SPM surface topography of unirradiated austenite and ferrite phases; and (**b**) nanoindentation load-displacement curves of the two phases in (**a**).

**Figure 3 micromachines-10-00011-f003:**
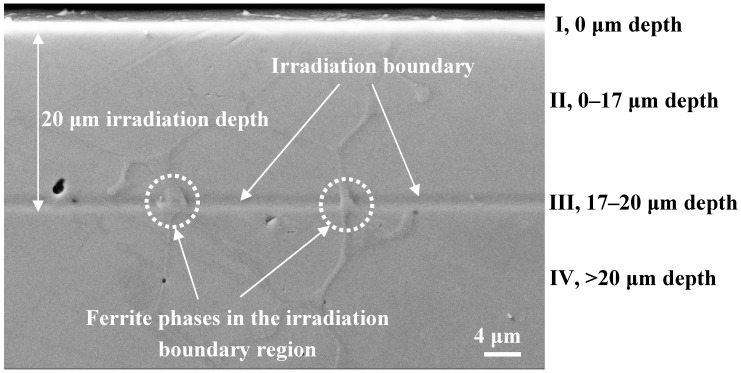
SEM image of the irradiated sample cross-section after colloidal silica slurry polishing including five regions: irradiated surface region I, uniform damage region II, maximum damage region III, and unirradiated region IV along the depth direction.

**Figure 4 micromachines-10-00011-f004:**
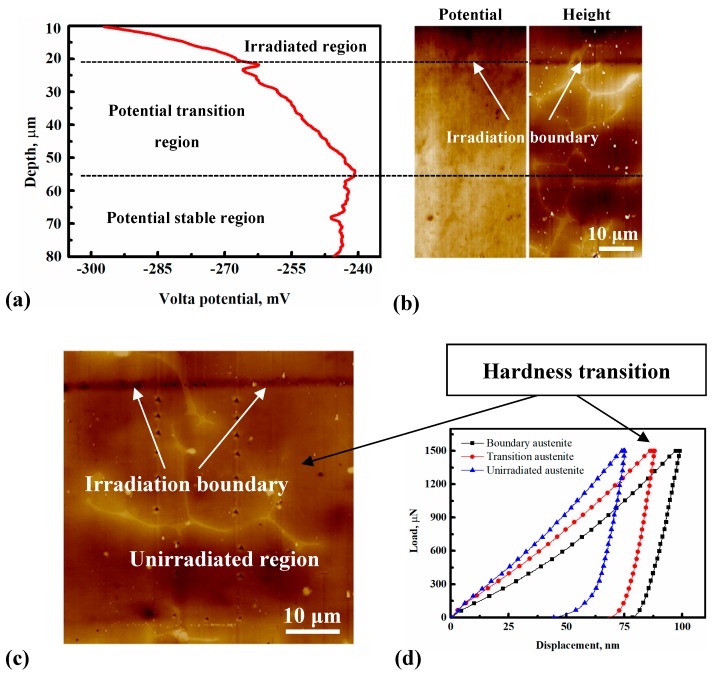
(**a**) The Volta potential curve vs. depth of the whole region shown in (**b**); (**b**) potential and topography height maps including irradiated (<20 μm depth) and unirradiated region IV along the depth direction; (**c**) AFM image showing all indents in the austenite γ phase in irradiation boundary (region III) and unirradiated region IV; and (**d**) nanoindentation load-displacement curves of the austenite γ phase in the irradiation boundary (region III), potential transition, and stable potential regions (>55 μm depth).

**Figure 5 micromachines-10-00011-f005:**
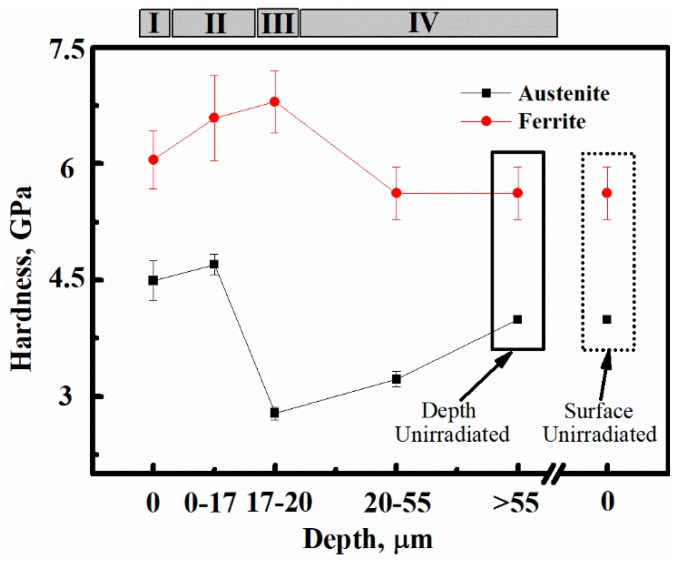
The hardness of the two phases in different regions: from the irradiated surface region I, uniform damage region II, maximum damage region III, to the unirradiated region IV along the depth direction, and the unirradiated surface and depth datapoints.

**Figure 6 micromachines-10-00011-f006:**
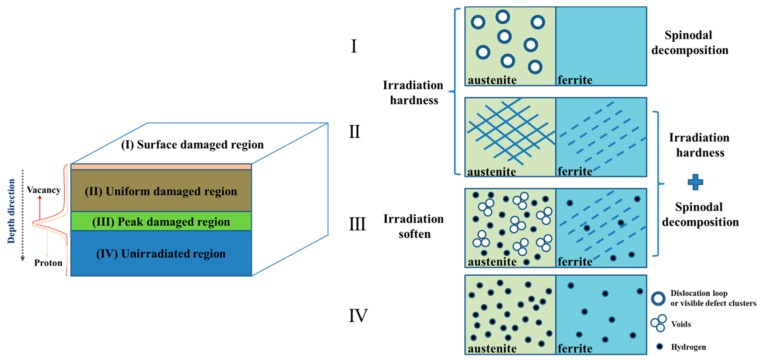
Schematic diagram showing how proton irradiation affects the hardness of austenite and ferrite in different regions.

**Table 1 micromachines-10-00011-t001:** Regions affected by vacancies and implanted protons.

Region Position in the Sample	Regions Affected by Vacancies and Protons	Depth, μm	Region Number
Surface	Surface damage region	0	I
Interior	Uniform damage region	0–17	II
	Maximum damage region	17–20	III
	Transition region (hardness and potential)	20–55	IV
	Region unaffected by irradiation	>55	

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
