# Peer review of "Proton Irradiation Effects on Hardness and the Volta Potential of Welding 308L Duplex Stainless Steel"

_micromachines, 2018, doi:10.3390/mi10010011_

Round 1

Reviewer 1 Report

This manuscript by Jiang et al. reports the difference between austenitic and ferritic stainless steels in hardening induced by 2-MeV proton irradiation at 633 K. The highlights of this article are the following two. The first is a discovery that the austenite exhibited abnormal softening at a depth (from the ion incident surface) corresponding to the maximum damage zone whereas the ferrite exhibited normal hardening in accordance with the damage level. The second highlight is that they have successfully mapped implanted protons by scanning Kelvin probe force microscopy (SKPFM). They demonstrated that protons are primarily located in a portion deeper than the maximum damage depth. Based on these results, the authors concluded that protons induced the softening of the austenite, though it remains an open question why not such softening occurred in the ferrite. I think their results are interesting enough to warrant publication in Micromechanics.

However, their way of paper writing is inappropriate. They have derived conclusions #1 and #2 not only from the facts, which they revealed by their experiments, but also from speculations. Despite they have not observed the microstructure, they discuss the effect of defect type on the hardening/softening behavior. They believe that the irradiation damage microstructure of their sample is the same as that of references #30 and #31. Those references performed microstructure characterization on samples subjected to heavy ion irradiation, whose resultant microstructure is in many cases largely different from that of light ion such as proton. As for the conclusion #3, although the authors discuss the difference between austenite and ferrite in terms of hydrogen solubility, it is also well known that hydrogen diffusivity is largely different. Hydrogen diffuses in ferrite much faster than in austenite [R1]. This may be a possible origin of the absence of hydrogen-induced softening in ferrite.

[R1] T. Mente and T. Bollinghaus Modeling of hydrogen distribution in a duplex stainless steel. Welding in the World 56, 66–78 (2012).

Author Response

Dear Reviewer,

Thank you so much for giving us the opportunity to revise this paper. We have carefully considered comments from you and have made revisions to address your concerns. More details about the revisions have been addressed in the word below.

Sincerely,

Baolong Jiang

Reviewer 2 Report

The manuscript is well written. I would recommend the manuscript to be accepted for publication.

Author Response

Dear Reviewer

I am also very grateful to you for your recommend about acceptance and publication of the manuscript.

Sincerely,

Baolong Jiang
